# Thin Benzotriazole Films for Inhibition of Carbon Steel Corrosion in Neutral Electrolytes

**Maxim Petrunin, Liudmila Maksaeva, Natalia Gladkikh, Yuriy Makarychev, Marina Maleeva, Tatyana Yurasova and Andrei Nazarov ***

Frumkin Institute of Physical Chemistry and Electrochemistry of the Russian Academy of Sciences, 119991 Moscow, Russia; mmvp@bk.ru (M.P.); lmaksaeva@mail.ru (L.M.); fuchsia32@bk.ru (N.G.); makarychev-1949@mail.ru (Y.M.); marina.maleeva@gmail.com (M.M.); tatal111@yandex.ru (T.Y.)

**\*** Correspondence: nazarovandrei@neuf.fr; Tel.: +33-298-456-241

**Abstract:** This article investigates the modification of a carbon steel surface by benzotriazole (BTA), and the structure and properties of the formed layers. Adsorption was studied by surface analytical methods such as X-ray photoelectron spectroscopy (XPS) and reflecting infrared microscopy (FTIR). It has been established that a polymer-like film containing iron-azole complexes that are 2 nm thick and strongly bonded to the metal is formed on the surface as a result of the azole interacting with a steel surface. This film is capable to inhibit uniform and localized corrosion of steel in neutral aqueous electrolytes containing chloride ions. It is shown that the iron-azole layer located at the interface acts as a promotor of adhesion, increasing the interaction of polymeric coatings with the steel surface. Taking into account these properties, the steel pretreatments can be used for improving the anticorrosion properties of polymeric coatings applied for the protection of steel constructions.

**Keywords:** corrosion inhibition; carbon steel; benzotriazole; X-ray photoelectron spectroscopy; adhesion

## 1. Introduction

The methods for corrosion protection of metallic constructions exposed to atmospheric or underground environments are continuously developing. These technologies have to prevent the environmental hazards caused by contamination with corrosion products of metals as well as toxic reagents and hydrocarbons that have evolved into the external environment upon violation of the integrity of chemical equipment and pipelines due to corrosion (penetrating corrosion defects) [1–4]. The application of polymeric coatings and corrosion inhibitors is most efficient for the protection from corrosion [5]. However, the adhesion of the polymeric coatings and the stability of the interface in corrosive environments decreases, leading to corrosion-induced deadhesion. To improve the stability, the surface pretreatments are normally applied. The most effective corrosion inhibitors and compositions for surface treatment contain hazardous and ecologically dangerous ions of hexavalent chromium ions, and now they cannot be applied. Thus, a significant amount of studies investigate the possible replacement of chromates in systems of corrosion protection.

Among the perspective inhibitors, organic heterocyclic compounds capable of forming stable complexes with cations of the metallic substrate are extremely important [6–17]. The resulting ultrathin (several nm thick) protective films are often formed, which are resistant to the impact of the corrosive environment (humid and polluted atmosphere, aqueous electrolytes of soil) [6,16]. The interest to heterocyclic compounds as corrosion inhibitors for ferrous and nonferrous metals has remained high due to their relatively low working concentrations, high reactivity, and protective efficiency [8–12]. Benzotriazole (BTA) (Figure 1) is a well-known corrosion inhibitor for copper and copper alloys that is most effective from heterocyclic compounds. Numerous works [6–9,17] have studied the corrosion

protective properties of BTA as well as its ability to form sparingly soluble compounds with cations of many metals (Ag, Cu, Zn, Fe, Ni, Co, and Pb ) [15,17].

Despite a significant number of studies on the inhibition by BTA, the information regarding the interaction of BTA with the surface of iron and/or steel is relatively rare. However, modification of steel surfaces by triazoles from the vapor phase and stable adsorption was proposed by the authors of [6]. BTA acts as a ligand in complexes with iron ions, becoming stable and sparingly soluble in water complexes (the log of stability constant of Fe (II) is 3.05 and of Fe (III) is 3.3 [12]). The stability of the complexes is the result of switching the spin state of a metal atom (the so-called "spin crossover") depending on external factors [18,19]. Moreover, elevated temperatures can cause a thermally induced transition between two electronic states in Fe (II) ions in these complex compounds: diamagnetic ($S = 0$, low-spin-LS) and paramagnetic ($S = 2$, high-spin-HS) [13]. Complex compounds of Fe(II) cations and substituted azole-containing compounds are often polymeric due to the bidentate nature of the azole ligand (Figure 2).

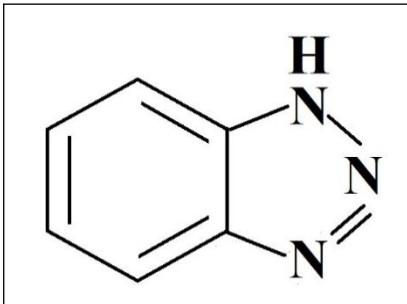

**Figure 1.** Structural formula of 1,2,3-benzotriazole.

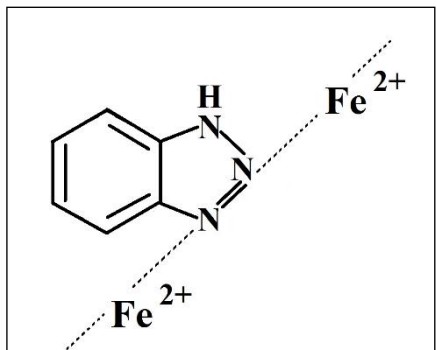
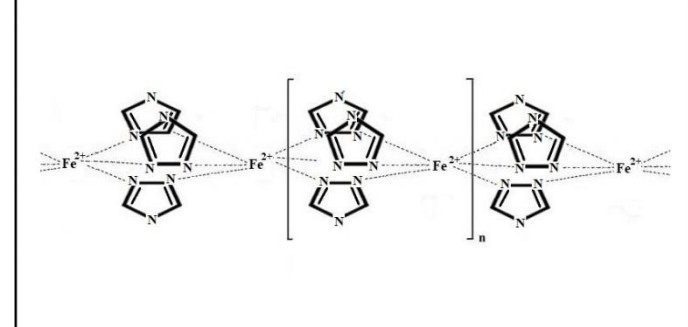

**Figure 2.** Schemes of the azole-containing compounds with $Fe^{2+}$ ions.

Corrosion inhibition of iron and steel will be provided by the surface layer formed upon BTA adsorption on the surface of the iron. However, in the literature, there is practically no information on the structure of the layers as a result of metal treatment with a solution containing BTA. A study of the adsorption of heterocyclic compounds would allow one to determine the properties of surface protective layers responsible for corrosion inhibition and to identify the advantages of heterocyclic compounds in comparison with other corrosion inhibitors [11,12]. The structure of the complexes present in Figure 2 estimates for species deposited from solution at a certain distance from the surface and cannot affect the metal corrosion significantly. The aim of this work is to study the structure and the properties of the grafted-to-surface BTA layers responsible for the inhibition of corrosion of steel. The work is part of the project devoted to the improvement of the anticorrosive properties of polymeric coatings for underground structures.

## 2. Materials and Methods

In this work, we used samples made of St3 carbon steel [20]. The composition is provided in Table 1. The following polymeric coatings were used in this work: decom bitumen-polymeric coating brand Decom (manufactured by "Delan" LLC, Russia) (BP) and coating (PC)-pentaphthalic paint of grade PF-15 (manufactured by "PKF Spectr" JSC, Russia). The samples of carbon steel had the shape of rectangular plates made of St3 steel. They were preliminarily grinded with 1000 grit ("Smirdex") emery paper, were degreased with acetone, and were immersed in the aqueous BTA solution. The solution contained 5 mM of 1,2,3-benzotriazole $C_6H_5N_3$ (manufactured by "ROD" LLC, Russia) (BTA). The modification time was 10 min. All the reagents were of "chemically pure" grade.

**Table 1.** Chemical composition of St3 carbon steel [21].

| Mass Fraction of Chemical Elements, % | | | | | | | |
|---|---|---|---|---|---|---|---|
| C | Mn | Si | Cr | Ni | Cu | S | P |
| 0.14–0.22 | 0.3–0.6 | <0.05 | <0.05 | <0.05 | <0.05 | <0.05 | <0.04 |

The electrochemical behavior of steel was studied by the DC polarization technique [22]. A three-electrode cell with separated electrode spaces was used. The potentials were measured using a saturated silver chloride reference electrode (Ag/AgCl) and were recalculated relatively to the standard hydrogen electrode (SHE) scale. The measurements were carried out using an IPC-Pro potentiostat (manufactured in Russia) at fixed potentials or potentiodynamically at a potential sweep rate of 0.1 mV/s. The experiments were performed on disk electrodes (area 0.8 cm$^2$) embedded into a plastic holder. The critical pitting potential ($E_{pt}$, Equation (1)) related to the potential of the pitting formation of the metallic substrate and propagation of stable pits [21]. It was determined from the anodic polarization curves by an inflection point on the curve at the potential, where a current increase was observed [21].

$$\Delta E_{pt} = E_{pt-BTA} - E_{pt-bg} \tag{1}$$

where $E_{pt-bg}$ and $E_{pt-BTA}$ are the pitting potentials without modification and after surface modification with a BTA solution, respectively [23]. In the electrochemical studies, the following borate buffer solution (pH 6.7) with the addition of chloride ions were used: 0.4 M $H_3BO_3$ + 0.1 M $Na_2B_4O_7$ + 0.01 M NaCl (background electrolyte).

Corrosion studies were carried out using accelerated corrosion tests in an MHK-408CL environmental chamber (manufactured in Taiwan), RH 95%, t = 60 °C). Rectangular samples (2 mm thick and area 9 cm$^2$) of St3 steel with a coating were tested. The coating thickness was controlled at 100 ± 5 μm using the thickness gauge. The corrosion damage of painted samples was estimated in accordance with the requirements of the international standard [24].

The adhesion of the bitumen-polymer coating was monitored by peeling the coating out from the metal substrate (dimensions 100 mm × 20 mm) at an angle of 180° in accordance with [25] on a Zwick–Roell Z 010 tensile testing machine (manufactured by Zwick GmbH & Co. KG, Ulm, Germany). The adhesive strength (A) in N/cm during peeling the coating was calculated using Equation (2):

$$A = \frac{F}{B} \tag{2}$$

where F-peeling force in a controlled area, and B-peeling strip width (2 cm). The second adhesion tests of the PC were carried out using the pull off test in accordance with [26]. The test samples were manufactured from St3 carbon steel as the discs with a diameter of 40 mm. A coating with the same thickness (100 ± 5 μm) was applied (Figure 3). After that, the sample and dolly were brought together and placed into a drying oven at T = 60 °C for 24 h until the coating hardened completely. To study the resistance of the adhesive joint to the action of corrosive atmospheric constituents, the samples were

kept for 10 days at 60 °C and at a relative humidity of 95% RH in an environmental chamber prior to testing.

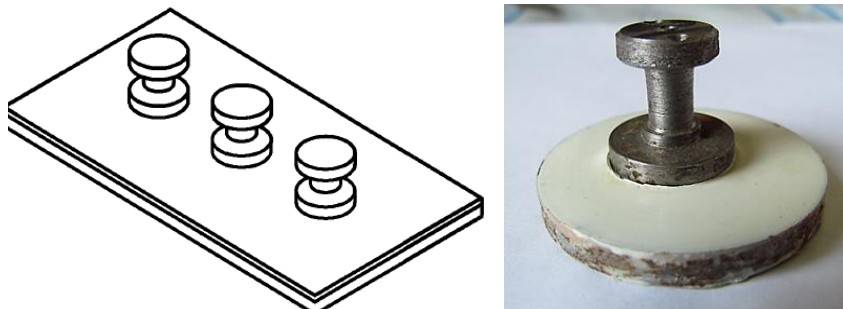

**Figure 3.** Samples for the adhesion test by dolly application.

To perform the adhesion tests, the larger disk was fastened in an immovable clamp, while a "dolly" was attached to the grippers of the tensile testing machine. The applied force required to pull off the "dolly" from the painted substrate was recorded using a tensile testing machine, and the adhesive strength A (MPa) was calculated using Equation (3) [26].

$$A = \frac{F}{S} \tag{3}$$

where F is the force required to pull off the dolly, and S is the pulled off area. All adhesion tests were carried in triplicates to get statistic data.

The chemical composition of the surface layers formed by the inhibitor on carbon steel samples was analyzed using XPS. An ESCALAB-5 X-ray spectrometer (manufactured by VG, Sussex, UK) was used for this purpose. The vacuum in the analytical chamber was $10^{-9}$ Torr. An Al anode with 200 W power was used as the excitation source. The pass energy of the analyzer was set to 50 eV. The distribution of chemical elements into the depth of samples was determined by etching with argon ions by energy 10 keV, a current density of 20 $\mu$A/cm$^2$, and a sputtering rate of 2.0 nm/min. Infrared spectra (FTIR) were recorded using the reflection mode of Hyperion 2000 IR (36× lens) microscope (Bruker Optic Gmbh, Ettlingen, Germany) connected to an IFS-66v/s spectrometer (Bruker) with resolution 2 cm$^{-1}$ in the range of 600–1500 cm$^{-1}$. The spectra were processed using the OPUS software package (Bruker Corporation). The Kramers–Kronig transform correction [27] was performed automatically.

## 3. Results

Figure 4 shows the XPS spectra of the steel surface after exposure of a sample in 5 mM aqueous BTA solution, and Figure 5 shows the N1s XPS spectra of BTA sodium salt and BTA molecules adsorbed on iron. The spectrum N1s of BTA sodium salt shows a symmetric peak with a binding energy of 399.6 eV and a peak half-width of 2.1 eV. The binding energy in the spectrum of BTA molecules adsorbed on iron is 400.6 eV, and the peak half-width is 2.6 eV.

To study the structure of the surface layer, an IR study of a steel surface modified with the BTA solution was carried out. Figure 6 shows the FTIR spectrum in the range of frequencies 600–1500 cm$^{-1}$. The IR spectrum contains bands that can be attributed to the components of the azole layer bound to the surface. In fact, the bands near 640 cm$^{-1}$ can be attributed to the vibrations of the triazole ring [19] and those near 1047 cm$^{-1}$ to the vibrations of the triazole and benzene rings (Figure 1) [29]. The bands at ca. 740–755 cm$^{-1}$, 885 cm$^{-1}$, and 1208 cm$^{-1}$ correspond to the vibrations of the CH bond in the benzene ring [19,28,29]. The bands at 1096 cm$^{-1}$ and 1201 cm$^{-1}$ correspond to the vibrations of the N–H moiety of the triazole group [14,22], while those at 995 cm$^{-1}$ and 1326 cm$^{-1}$ correspond to the vibrations of the N–N bonds in the triazole ring [29–31]. The bands at 1347 cm$^{-1}$, 1351 cm$^{-1}$, and 1380 cm$^{-1}$ correspond to the vibrations of the –C–N bond in the triazole moiety [29], while the band at

1590 cm$^{-1}$ corresponds to the –C=C vibrations of the benzene ring [24]. Moreover, the bands at about 1090, 1105, 1400, and 1415 cm$^{-1}$ are attributed to complexes of azoles with Fe(II) [32], while the bands at 623, 1490, and 1500 cm$^{-1}$ can be attributed to the vibrations of the Fe–N bonds (Figure 2) [19].

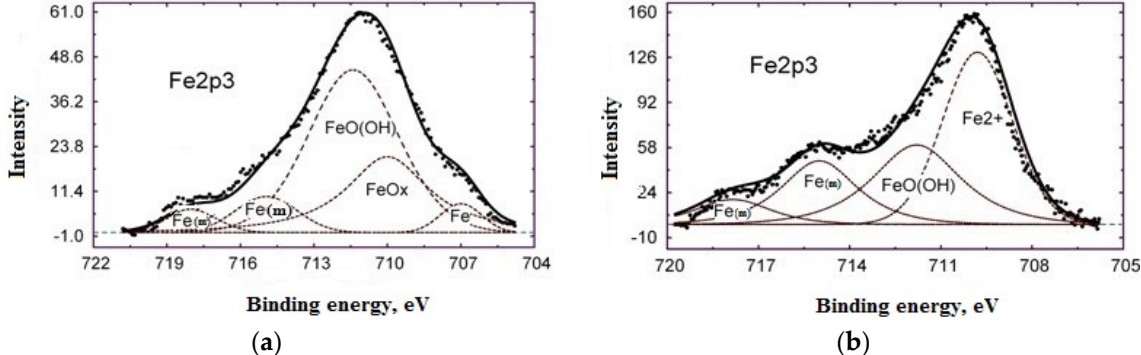

(**a**)　　　　　　　　　　　　　　　　　　　　　　　　(**b**)

**Figure 4.** (**a**) XPS spectrum of Fe 2p$^{3/2}$ electrons of steel surface: an after exposure of steel samples in the background electrolyte (10 min); (**b**) after exposure of steel samples in the background electrolyte with the addition of 5mM benzotriazole (BTA) (duration 10 min). The chemical species are given in the spectra. Fe (m) – metallic iron.

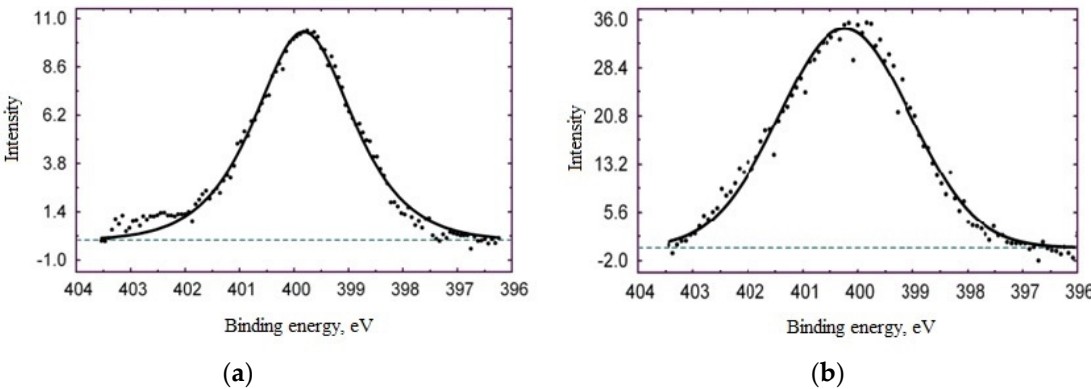

(**a**)　　　　　　　　　　　　　　　　　　　　　　　　(**b**)

**Figure 5.** XPS N1s spectra of BTA sodium salt (**a**) and BTA molecules adsorbed on iron (**b**).

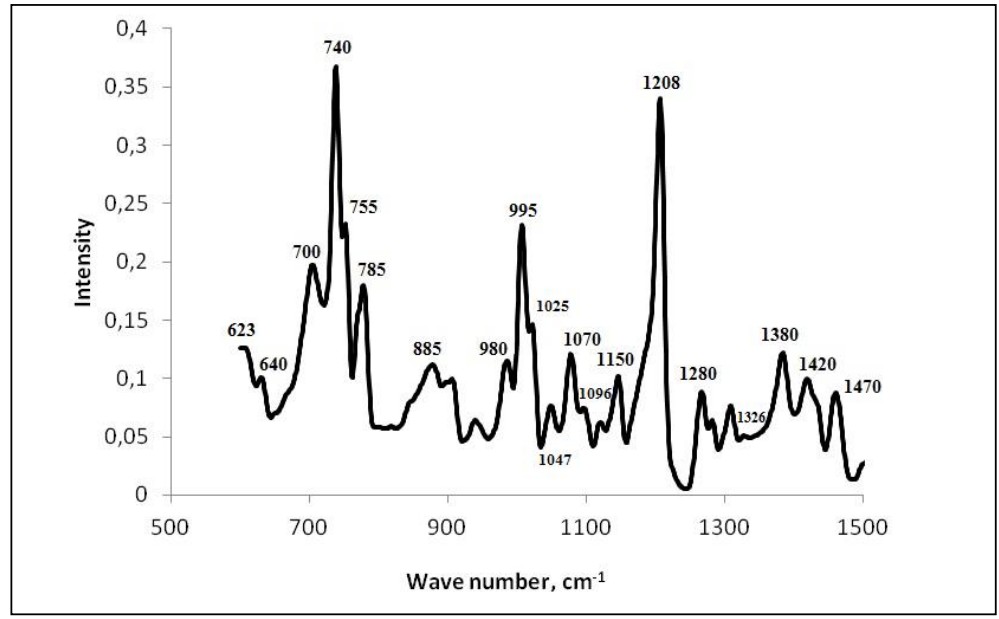

**Figure 6.** FTIR spectrum of the surface of carbon steel modified in 5 mM aqueous BTA solution.

The effect of surface benzotriazole-containing layers on the electrochemical behavior of the metal was studied. Figure 7 demonstrates the anodic polarization curves measured from open circuit potential (see experimental part). The data are in line with the vapor-modified BTA steel [6].

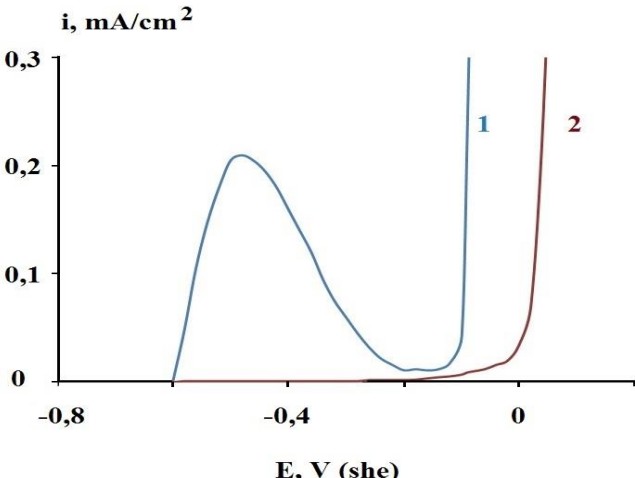

**Figure 7.** Anodic potentiodynamic polarization curves of St3 carbon steel. 1. Nonmodified steel; 2. steel modified with 5 mM aqueous solution of BTA. Borate buffer solution with the addition of 0.01 M NaCl, pH 6.7. Potential sweep rate: 0.1 mV/s.

Corrosion and adhesion studies were carried out in order to estimate the effect of the benzotriazole surface layer on the anticorrosion properties of the coating. Figure 8 shows the results of the measurements of the adhesive strength of a bitumen-polymer coating (BP) and pentaphtalic paint coating (PC) applied on a nonmodified steel surface and to a sample modified with a BTA solution. The samples were kept for 10 days at 60 °C and at a relative humidity of 95% RH in an environmental chamber prior to testing. The pull off setup was applied.

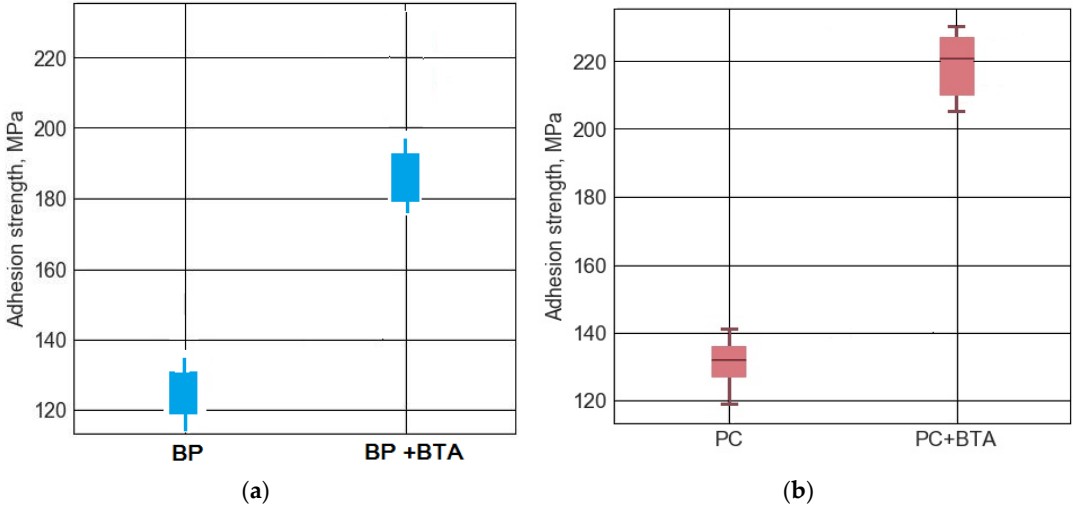

**Figure 8.** Strength of the adhesion for metal (steel)-polymer coatings (BP and PCA) for nonmodified and modified (BP+BTA) and (PC+ BTA) steel surfaces: (**a**) bitumen-polymeric coating (BP); (**b**) pentaphtalic paint coating (PC). Error bars correspond to measurements of three specimens.

The surface of the samples after peeling tests is shown in Figure 9. When the coating was peeled out from nonmodified metal, the locus was of mixed type but mostly adhesive, i.e., the coating was detached from the metal surface (Figure 9a). In the case of the preliminary modification of the substrate surface with the BTA solution, cohesive failure occurred (Figure 9b), i.e., tear-off involved destruction

inside the polymer layer. This indicates an increased "true" adhesion in the presence of a benzotriazole layer on the steel surface.

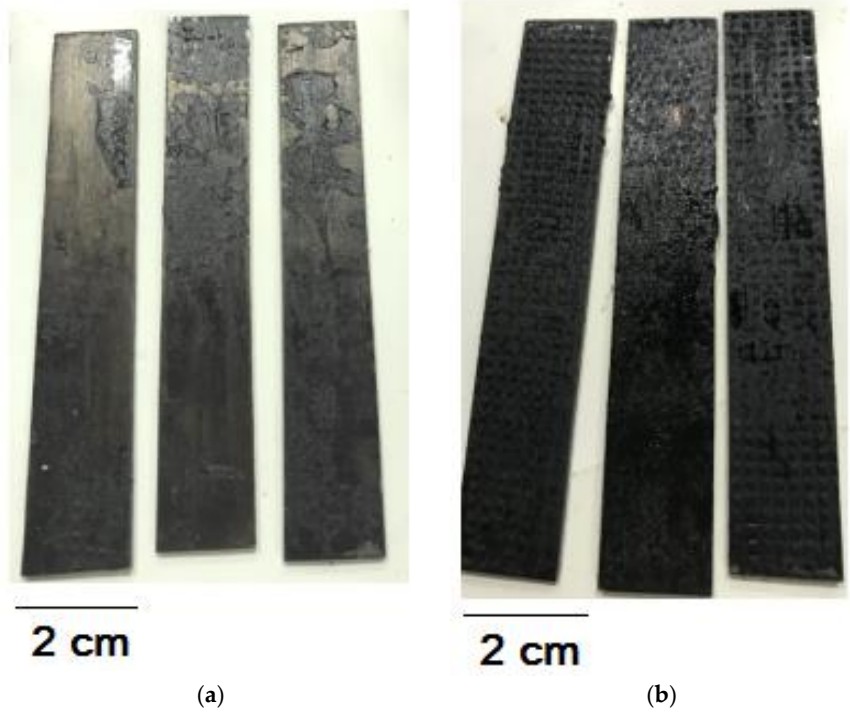

**Figure 9.** Photographs of metal-polymer (BP) adhesive joints: (**a**) nonmodified steel, (**b**) steel modified with the BTA solution. The sample dimensions are 100 mm × 20 mm.

The effect of the steel surface pretreatments by BTA on the corrosion of steel with applied pentaphthalic paint coating (PC) was studied. For this purpose, corrosion tests of steel samples coated with the PC were carried out, after which the coating adhesion was determined. Figure 10 shows the appearance of the samples after 10-day tests in the environmental chamber. Inhibition of the under film corrosion in the presence of a surface benzotriazole layer can be observed: the surface area affected by corrosion estimated in accordance with [13] was 24% and 5% for nonmodified and BTA-modified surfaces, respectively. Adhesion measurements showed a significant increase in adhesion for the modified samples: the adhesive strength was 134 and 221 MPa for nonmodified and modified samples, respectively (Figure 8b).

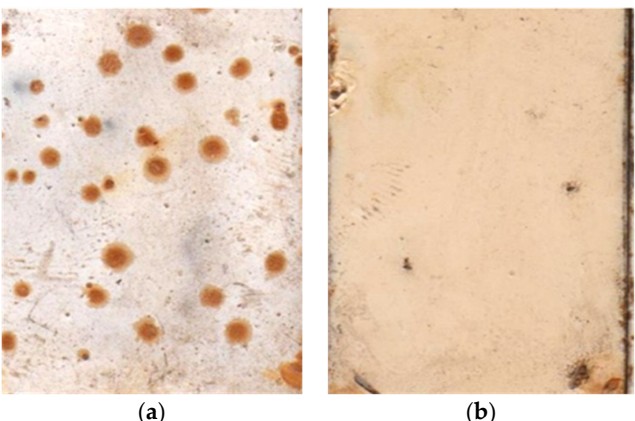

**Figure 10.** Samples of carbon steel coated with the PC after corrosion tests in the environmental chamber. Ten days. T 60 °C, RH 95%: (**a**) nonmodified steel, (**b**) steel premodified with the BTA solution. Coating thickness – 0.1 mm.

## 4. Discussion

It is known that upon adsorption on iron, molecules of triazoles (in particular, BTA) can form chemisorption layers [28] as well as polymer-like complexes $Fe_n$-$BTA_m$. In both cases, the structure and thickness of the adsorbed layers were determined from $Fe2p^{3/2}$ and N1s XPS spectra. Analysis of the XPS spectra (Figures 4 and 5) shows that the presence of peak's corresponding metallic iron and FeOOH on Fe3p2 spectrum indicates that the thickness of the surface film does not exceed 2–3 nm. It is in line with data of ellipsometry for vapor-adsorbed BTA [6].

The sN1 spectrum of BTA sodium salt shows a symmetric peak with a binding energy of 399.6 eV and a peak half-width of 2.1 eV. The binding energy in the spectrum of BTA molecules adsorbed on iron is 400.6 eV, and the peak half-width is 2.6 eV. A comparison of these spectra can indicate the bonding between BTA molecules and iron [28]. A shift in the binding energy maximum towards higher energies and an increase in the peak half-width are observed due to the chemisorption nature of the layer. Similar results were obtained in [28] in a study of the adsorption of 2-mercaptobenzothiazole (2-MBT) on iron, where it was found that the chemical interaction of 2-MBT molecules with iron occurs due to the formation of a donor-acceptor bond between the lone electron pair on nitrogen atoms of the azole ring and the free d-orbital of the iron atoms of the substrate.

Immersion of samples of steel in a buffer solution at pH 6.7 leads to the formation of a very thin film of oxides on the surface (probably FeOOH and $Fe_3O_4$ $E_{sv}$ = 711.6 and 710.6 eV for Fe2p3 2, Esv = 56.6 and 55.2 eV for Fe3p electrons, respectively) (Figure 4a), since a peak is observed due to metallic iron. These values are in good agreement with published data [8]. The surface is noticeably hydroxylated, since there is a peak at 532 eV, which is obtained by the decomposition of the oxygen spectrum along with a peak of oxygen at 530 eV, which is part of the structure of iron oxide. Using the integrated intensities of the peaks of the corresponding elements that make up the surface layers leads to the following layer thicknesses: carbon pollution -0.8 nm, OH-1.2 nm, and iron oxides 1.8 nm [11].

The exposure of steel samples in a borate solution containing 5 mM BTA at pH 7.36 leads to the appearance of an N1s peak of electrons and the disappearance of a peak of metallic iron (Figure 5). The Fe2p spectrum consists of two doublets due to $Fe^{3+}$ and satellite peaks shifted by 8 eV. The peaks of Fe $2p^{3/2}$ at 711.6 eV and Fe3p at 56.5 eV correspond to the presence of FeOOH in the surface film. The position of the symmetric N1s peak of electrons at 400.7 eV does not correspond to the value observed for the salt BTA–Na. Note that the peak width $A_{1/2}$ is <2.0 eV, which indicates a uniform distribution of electron density over three nitrogen atoms.

Based on the assumption that BTA is adsorbed without complex formation, the calculated thickness of the BTA layer on the FeOOH surface is an average of ~ 0.35 nm. Insertion of iron ions to the composition of the surface film in the ratio Fe: BTA = 1:1 or 1:2 does not lead to significant changes in the calculated layer thickness. In the first case, the contribution of the intensity of the iron cations included in the complex does not exceed 10%. With a small change in $E_{cb}$ for Fe3p in the case of complex formation (i.e., a change in the electron charge on the iron cation upon transition from FeOOH to Fe: BTA), it is difficult to expect noticeable differences in the spectrum of Fe3p. Replacing oxygen as a ligand with a nitrogen atom does not lead to noticeable changes in the charge on the iron cation, which cannot affect the overall spectrum. We note here that, in contrast to a rather thin oxide film (1.8 nm) (Figure 4a.), which is observed when steel is held in the background solution, in the case of the presence of BTA in the film, the iron oxide film is thicker. Based on the ion etching of the surface in the spectrometer chamber, the thickness of the oxide layer reached 6–8 nm, on which there is a grafted BTA layer of 0.35 nm thick or more.

An analysis of the IR spectra showed the presence of both azole groups and complexes of azoles with iron ions on the metal surface. Thus, pretreatment of the carbon steel surface with the aqueous BTA solution results in the formation of iron-azole polymer-like surface layers strongly bound to the oxide layer of the steel.

The structure of this layer is shown in Figure 11. It is similar to that of complexes presented in [33–35]. A calculation of the possible layer thickness based on the bond lengths and comparison with

the thickness obtained from XPS measurements showed that the value of *n* close to 4. It means that the thickness of the iron-azole surface film is four molecular layers or 2 nm. Such layers are capable of inhibiting metal corrosion [6,13,17]. In combination with organosilanes, they can increase the adhesion of an anticorrosive paint coating [34]. The studies of the electrochemical behavior of steel (Figure 7) show that the presence of an iron-benzotriazole polymer-like layer increased the potential of pitting formation and vanished the peak corresponding to the active-passive transition on the anodic curve that indicates that passivation of the metal occurs more easily in the presence of the surface BTA layer. Thus, the results may indicate that the iron-benzotriazole surface layer can inhibit both uniform and local steel corrosion [6,35].

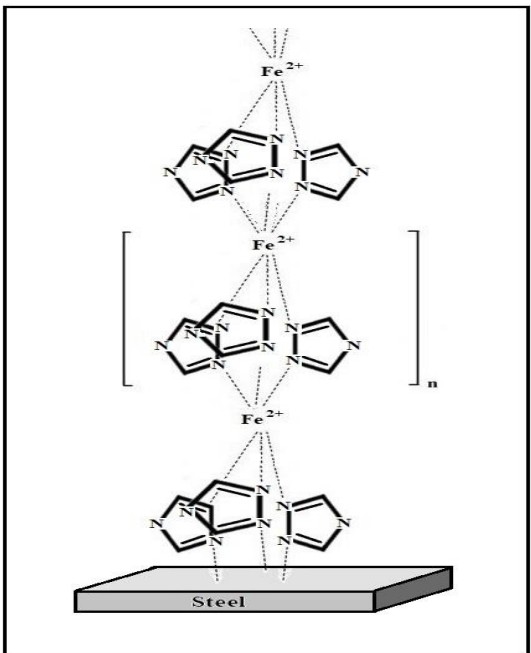

**Figure 11.** Schematic of the structure of the polymer-like iron-azole surface layer.

It has been shown that the treatment of steel surface increases the corrosion stability of steel-coated by polymeric coatings, since it can both increase the adhesion of the coating to the metal and reduce the rate of under film corrosion of steel. Thus, it is possible to suppose that a three-dimensional polymer-like layer is working as an adhesion promoter and limits the access of aqueous electrolyte to the coating-metal interface.

## 5. Conclusions

- A study on the adsorption of BTA on steel surface using XPS and IR spectroscopy showed that modification of carbon steel by an aqueous BTA solution produces a polymer-like iron-azole film with a thickness of four molecular layers (thickness 2 nm). The layer contains a bridged complex with donor-acceptor Fe-triazole bonds;
- The layer is capable of inhibiting both uniform and local corrosion of steel in the presence of chloride ions;
- Pretreatment of steel surface with a benzotriazole solution improves the anticorrosion characteristics such as the adhesion of the coating to the steel. The improvement of the adhesion was shown by peel out and pull off tests of the coatings after exposure in high humid air at the elevated temperature. The corrosion test shows that undercoating corrosion is also reduced by preliminary grafting to the steel surface benzotriazole layer, acting as a corrosion inhibitor and adhesion promoter.

**Author Contributions:** Conceptualization (M.P.); writing of the article (M.P., L.M., A.N.), design of experiments (N.G., Y.M., T.Y.), surface analytical investigations (Y.M., M.M.), English editing (A.N.). All authors have read and agreed to the published version of the manuscript.

**Funding:** This research was funded by the BASIC RESEARCH PROGRAM OF THE PRESIDIUM OF THE RUSSIAN ACADEMY OF SCIENCES, "Urgent Problems of Surface Physical Chemistry and Creation of New Composite Materials. Nanostructured Coatings for Electronics, Photonics, Alternative Energy Sources, and Materials Protection". The grant number is 24P.

**Conflicts of Interest:** The authors declare no conflict of interest.

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
