# Peer review of "Thin Benzotriazole Films for Inhibition of Carbon Steel Corrosion in Neutral Electrolytes"

_coatings, doi:10.3390/coatings10040362_

Round 1
Reviewer 1 Report
- Introduction.
Please define thermodynamic stability of the complexes and solubility in water, possibility of complex formation with Fe(III) species. - Line 107 Please equation must be written using MATH editor of Word and it is not just picture. It is related to other equations in the text.
- 145 Can you give also deviations from mean value of peel strength or pull off strength?
- Figure 4, please add the survey XPS spectrum.
- 177 Please add the literature data about chemical shift in XPS for other different complexes to support your data.
- Figure 6. You show only part of FTIR spectrum. Please give the survey spectrum. Above this part of spectrum you can add theoretical lines according to cited literature data.
- Figure 8, X -axe give more fine scaling and replace nhe to SHE.
- 230 please give the data about how many samples were used and deviations (is it mean data?).
- Figure 9, how many samples were used (triplicates?) give the deviation bar.
- 249 what it is “true” adhesion, probably it is cohesion strength of the material.
- Figure 10. Were the samples exposed in some kind of test (on the non-treated surface rust is visible)?
- Add this data to Figure caption.
- 260 Give the data about accelerated corrosion test: humidity temperature, addition of the salt, etc.
- Figure 11. What is the coating thickness?
Author Response
Dear Mr. / Mrs. Reviewers, Editors, thank you very much for your work with the article. No doubts your comments improved the manuscript significantly.
Introduction.
Please define thermodynamic stability of the complexes and solubility in water, possibility of complex formation with Fe(III) species.
Thank you very much. The constants of the stability with Fe(II) and Fe(III) and data about solubility were inserted in Introduction part.
Line 107 Please equation must be written using MATH editor of Word and it is not just picture. It is related to other equations in the text.
Thank you very much. It was done.
145 Can you give also deviations from mean value of peel strength or pull off strength?
Thank you very much. It was added as error bar to the Figure 8.
Figure 4, please add the survey XPS spectrum.
Thank you very much. The survey spectrum was discussed in the text.
177 Please add the literature data about chemical shift in XPS for other different complexes to support your data.
Thank you very much. The comparison with literature was added to the text.
Figure 6. You show only part of FTIR spectrum. Please give the survey spectrum. Above this part of spectrum you can add theoretical lines according to cited literature data.
Thank you very much. The measurements were performed in particular frequency range that is informative of BTA adsorption.
Figure 8, X -axe give more fine scaling and replace nhe to SHE.
Thank you very much. It is done.
230 please give the data about how many samples were used and deviations (is it mean data?).
Figure 9, how many samples were used (triplicates?) give the deviation bar.
Thank you very much. The error bar was added to the figures. In the caption was noted about triplicate.
249 what it is “true” adhesion, probably it is cohesion strength of the material.
Thank you very much. The explanation was added to the text.
Figure 10. Were the samples exposed in some kind of test (on the non-treated surface rust is visible)? Add this data to Figure caption.
Thank you very much. It was done.
260 Give the data about accelerated corrosion test: humidity temperature, addition of the salt, etc.
Figure 11. What is the coating thickness?
Thank you very much. The thickness 0.1mm, the data were added to the text.
Reviewer 2 Report
This is a good paper about an industrially and socially important problem (corrosion of steel construction materials).
The analytical work is good but there are several issues that must be addressed before the paper could be published:
- The English is awkward (incorrect word choices and phrasing) in the "Introduction" section. The other parts of the paper need only minor editing.
- Figures 6 & 8 have no axes on the graphs. It is difficult to understand which peaks are being discussed when the results in Figure 6 are explained in the text without some point of reference on the axes. Although the curves shown in Figure 8 indicate the efficacy of the BTA treatment, the figure would be clearer if there were axes.
- In Figure 9, although the caption is clear, the bar graphs themselves are not adequately identified and there is no quantitative y-axis to understand the magnitude of improved adhesion.
Author Response
Dear Mr. / Mrs. Reviewers, Editors, thank you very much for your work with the article. No doubts your comments improved the manuscript significantly.
This is a good paper about an industrially and socially important problem (corrosion of steel construction materials). The analytical work is good but there are several issues that must be addressed before the paper could be published:
The English is awkward (incorrect word choices and phrasing) in the "Introduction" section. The other parts of the paper need only minor editing.
Thank you very much. Additional English text edition was done.
Figures 6 & 8 have no axes on the graphs. It is difficult to understand which peaks are being discussed when the results in Figure 6 are explained in the text without some point of reference on the axes. Although the curves shown in Figure 8 indicate the efficacy of the BTA treatment, the figure would be clearer if there were axes.
Thank you very much. The axis were added to the Figures to clarify the points.
In Figure 9, although the caption is clear, the bar graphs themselves are not adequately identified and there is no quantitative y-axis to understand the magnitude of improved adhesion.
Thank you very much. The dimensions of the samples were added to the Figure capture.
Reviewer 3 Report
This manuscript of Nazarov et al. is written fairly well, but there are some important issues to be solved related with the experimental data and their interpretation.
Thus, the novelty of this work must be better highlighted both in introduction and in experimental sections of the manuscript. This aspect is very important since there are other papers in the literature, cited or not in this study, which refer to the use of 1,2,3-Benzotriazole (BTA) as corrosion inhibition agent used to protect steel, or to study the interaction of BTA with the surface of iron and/or steel. I found in the literature a recent study (Int. J. Corros. Scale Inhib., 2018, 7, no. 2, 203–212, doi: 10.17675/2305-6894-2018-7-2-7) on the same topic as the present one, which was not cited in the manuscript, and I kindly ask the authors to highlight the differences between the two studies regarding the study of the interaction between BTA and steel? What does this study add to the previous ones? This aspect should be found in the introduction.
The references are close enough and are presented in the appropriate format for Coatings. The entire manuscript respects the journal's requirements.
However, there are some aspects that make the manuscript difficult to read and understand. Here are some suggestions for the authors:
- The first phrase in the Introduction should be revised. There are two words with capitals there.
- Page 2, line 56 - Authors stated that: “Numerous works [11] deal with the studies of the protective 56 properties of BTA due to its commercial availability and good solubility of BTA in water,…” but they provided only one citation here.
- Please provide schemes with increase resolution for Figures 1,2 and 7!
- The quality of all figures presented here should be improved. The numbers and the text in the figures and even axes are missing (see Figures 6, 8 and 9).
- Figure 4: More details should be provided in the figure caption. All the curves should be explained.
- Figure 5: My opinion is that this figure would be clearer if the same scale for intensity were used in both figures.
- Figure 6 this should be replaced with one providing more details (peak position, wave numbers, etc). Here I can see only a FT-IR spectra, without any numbers on the OX and OY axes.
- More details should be introduced in Conclusion section.
Author Response
Dear Mr. / Mrs. Reviewers, Editors, thank you very much for your work with the article. No doubts your comments improved the manuscript significantly.
This manuscript of Nazarov et al. is written fairly well, but there are some important issues to be solved related with the experimental data and their interpretation.
Thus, the novelty of this work must be better highlighted both in introduction and in experimental sections of the manuscript. This aspect is very important since there are other papers in the literature, cited or not in this study, which refer to the use of 1,2,3-Benzotriazole (BTA) as corrosion inhibition agent used to protect steel, or to study the interaction of BTA with the surface of iron and/or steel. I found in the literature a recent study (Int. J. Corros. Scale Inhib., 2018, 7, no. 2, 203–212, doi: 10.17675/2305-6894-2018-7-2-7) on the same topic as the present one, which was not cited in the manuscript, and I kindly ask the authors to highlight the differences between the two studies regarding the study of the interaction between BTA and steel? What does this study add to the previous ones? This aspect should be found in the introduction.
Thank you very much. The reference was added to the list of literature and was discussed in Introduction and discussion parts. In fact the setups of two articles is very different. In the literature the absorption was carried out from vapor phase and investigations of the formed layer were performed using ellipsometry and corrosion test. In our article were used deposition from aqueous electrolyte and XPS and FTIR were used to study the formed layers. Important point of our works is formulation of adhesion joints and conclusion about work of BTA layers as adhesion promoters.
The references are close enough and are presented in the appropriate format for Coatings. The entire manuscript respects the journal's requirements.
However, there are some aspects that make the manuscript difficult to read and understand. Here are some suggestions for the authors:
The first phrase in the Introduction should be revised. There are two words with capitals there.
Thank you very much. It was done.
Page 2, line 56 - Authors stated that: “Numerous works [11] deal with the studies of the protective 56 properties of BTA due to its commercial availability and good solubility of BTA in water,…” but they provided only one citation here.
Thank you very much. The text was improved and more related to subject references were added.
Please provide schemes with increase resolution for Figures 1,2 and 7!
The quality of all figures presented here should be improved. The numbers and the text in the figures and even axes are missing (see Figures 6, 8 and 9).
Thank you very much. The axes were added and the Figures quality was improved.
Figure 4: More details should be provided in the figure caption. All the curves should be explained.
Thank you very much. The spectra were explained in the text.
Figure 5: My opinion is that this figure would be clearer if the same scale for intensity were used in both figures.
Thank you very much. We try to prepare new scales but it leads to loosing of the resolution of the spectra.
Figure 6 this should be replaced with one providing more details (peak position, wave numbers, etc).
Here I can see only a FT-IR spectra, without any numbers on the OX and OY axes.
Thank you very much. The axes were improved.
More details should be introduced in Conclusion section.
Thank you very much. The conclusions part was improved and additional information was added.
Reviewer 4 Report
In this work, authors investigated the structure and composition of the layers of benzotriazole after modification of the steel surface, aiming to present solution to protect underground steel constructions. The work is interesting and reported results are of great importance for researchers working on this field. After reading the entire manuscript, it seems that there are only some points should be considered before its acceptation for publication in Coatings. These points are:
- “The work is a part of the project for improvement of the corrosion stability of underground steel constructions by using polymeric coating containing corrosion inhibitors”. It seems not appropriate for abstract.
- Results and discussion should be organized in separate sections. It’ll make the manuscript more readable.
- “Figure 6 shows the FT-IR spectrum in the range of frequencies 500-1500 cm-1”. Why this included in caption of Figure 5.
- Figure 6 should be improved. It’s without axes.
- Same for Figure 8.
- Figure 9 is not useful. Authors should change it or present it correctly.
- Authors should present a more detailed mechanism in a separate section.
- Please be sure that your Conclusions section not only summarizes the key findings of your work but also explains the specific ways in which this work fundamentally advances the field relative to prior literature.
Author Response
Dear Mr. / Mrs. Reviewers, Editors, thank you very much for your work with the article. No doubts your comments improved the manuscript significantly.
In this work, authors investigated the structure and composition of the layers of benzotriazole after modification of the steel surface, aiming to present solution to protect underground steel constructions. The work is interesting and reported results are of great importance for researchers working on this field. After reading the entire manuscript, it seems that there are only some points should be considered before its acceptation for publication in Coatings. These points are:
“The work is a part of the project for improvement of the corrosion stability of underground steel constructions by using polymeric coating containing corrosion inhibitors”. It seems not appropriate for abstract.
Thank you very much. The phrase was removed from the abstract.
Results and discussion should be organized in separate sections. It’ll make the manuscript more readable.
Thank you very much. The re-structuring of the manuscript according to comment was done.
“Figure 6 shows the FT-IR spectrum in the range of frequencies 500-1500 cm-1”. Why this included in caption of Figure 5.
Thank you very much. The mistake was found and the captions re-phrased.
Figure 6 should be improved. It’s without axes.
Thank you very much. The axes were added to the Figure.
Same for Figure 8.
Thank you very much. The axes were added to the Figure.
Figure 9 is not useful. Authors should change it or present it correctly.
Thank you very much. I agree the Figure has very qualitative information and just proves the under film corrosion inhibition. We added the dimensions bar.
Authors should present a more detailed mechanism in a separate section.
Please be sure that your Conclusions section not only summarizes the key findings of your work but also explains the specific ways in which this work fundamentally advances the field relative to prior literature.
Thank you very much. The novelty of the work was highlighted in the conclusion part.
Round 2
Reviewer 1 Report
The article was improved and it is ready for publication. I recommend to publish the article as it is.
Reviewer 3 Report
I have read the answer carefully and I agree with the publication of the study in this form.